# Solvent-Free Processed Cathode Slurry with Carbon Nanotube Conductors for Li-Ion Batteries

**DOI:** 10.3390/nano13020324

**Published:** 2023-01-12

**Authors:** Gyori Park, Hyun-Suk Kim, Kyung Jin Lee

**Affiliations:** 1Department of Chemical Engineering and Applied Chemistry, Chungnam National University, Daejeon 34134, Republic of Korea; 2Department of Materials Science and Engineering, Chungnam National University, Daejeon 34134, Republic of Korea

**Keywords:** Li ion battery, dry process, solvent free electrode, carbon nanotube, NCM811

## Abstract

The increase in demand for energy storage devices, including portable electronic devices, electronic mobile devices, and energy storage systems, has led to substantial growth in the market for Li-ion batteries (LiB). However, the resulting environmental concerns from the waste of LiB and pollutants from the manufacturing process have attracted considerable attention. In particular, N-methylpyrrolidone, which is utilized during the manufacturing process for preparing cathode or anode slurries, is a toxic organic pollutant. Therefore, the dry-based process for electrodes is of special interest nowadays. Herein, we report the fabrication of a cathode by a mortar-based dry process using NCM811, a carbon conductor, and poly(tetrafluoroethylene)binder. The electrochemical performance of the cathode was compared in terms of the types of conductors: carbon nanotubes and carbon black. The electrodes with carbon nanotubes showed an ameliorated performance in terms of cycle testing, capacity retention, and mechanical properties.

## 1. Introduction

Recently, owing to the demand for energy storage systems, including electric car batteries, the requirement for the production of secondary batteries, particularly, Li-ion batteries (LiB) is significantly high [1,2,3,4,5,6]. In addition to the advancements in the search for alternatives for energy storage systems, research has been devoted to developing high-performance LiBs with better safety and lifetime [7,8,9,10,11,12]. Simultaneously, there has been considerable attention on environmental issues potentially arising from the waste or manufacturing procedures of LiB [13,14,15,16,17,18,19]. During the fabrication of cathodes or anodes, most industries have utilized N-methylpyrrolidone (NMP)-based slurry containing active materials, conductors, binders, and several additives [20,21,22,23]. However, owing to factory safety and the generated pollutants, environmental regulations in the near future concerning organic solvents will lead to new approaches to preparing LiB components [24]. Several excellent methods have been reported, including water-based electrode slurries. Rongyu et al. and Minh et al. reported water-soluble binders for anode electrodes in LIBs [25,26]. Weiwen et al. also reported water-based binders for cathode electrodes in Li-S batteries [27]. However, adopting an aqueous-based slurry on the cathode electrode results in limitations because most of the active materials in the cathode are unstable under water and humidity exposure. In addition, wet processing such as NMP and aqueous-based slurry methods require a long processing time and cost because the solvent must be sufficiently dried, and microstructural defects may occur on the electrode surface while the solvent evaporates. Therefore, dry processes to fabricate cathodes, including the dry jet and mortar methods, which were recently reported, is essential. Mohanad et al. reported a solvent-free dry-powder coating process for NCM-positive electrodes in LIBs. Dylan et al. reported a dry pressing method for binder-free Li-ion electrodes, and Ludwiget et al. reported a dry powder painting process of electrodes for LIB [28,29,30]. In addition, the agate mortar method has been adopted to disperse active materials and conductors into binders; however, limited information has been provided so far [31,32].

In this study, we report a dry-processed LiB cathode composed of NCM811, poly(tetrafluoroethylene) (PTFE) binders, and carbon conductors. We compared the electrochemical performances in terms of the types of conductors: carbon nanotubes (CNT) and carbon black (CB). All components in the cathodes are effectively dispersed in the PTFE binder using the agate mortar method without the addition of an organic solvent. Since the solvent removal process is not required, the dry process using the PTFE binder can reduce the processing time and does not result in unnecessary voids on the electrode surface. The density of the cathode is slightly higher in the case of the CNT conductor than in the case of the CB conductor, resulting in enhanced electrochemical performance, including overall capacity, durability, and cycle performance. The results of this study can provide basic guidelines for future battery designers to consider solvent-free processes. In addition, these solvent-free processes could be effectively utilized in the fabrication of all-solid-state LiB.

## 2. Materials and Methods

### 2.1. Materials

CNT (JEIO, Republic of Korea, 5~7 nm) and carbon black Super P (Alfa Aesar, Conductive 99+%, 40 nm) were used as conductive additives in this study. LiNi_0.8_Co_0.1_Mn_0.1_O_2_ (NCM811) (Wellcos, Republic of Korea, 7~14 μm) was used as the cathode material, and polytetrafluoroethylene (PTFE) (Chemours-Mitsui Fluoroproducts Co.,Tokyo Ltd., Japan, 495 μm) was used as the binder without further purification.

Scanning electron microscopy (SEM) (TESCAN. CLARA) was used to investigate the microstructure of the cathode before and after cycling. The mechanical properties of the cathode with a thickness of approximately 250 μm were analyzed using a universal testing machine (UTM; Lloyd LRX Plus, AMETEK). Electrochemical impedance spectroscopy (EIS, Ivium-n-stat) was used to perform impedance analysis, and a battery testing system (WBCS 3000; WonAtech, Seoul, Republic of Korea) was used to analyze the cycle performance and discharge C-rate capacity.

### 2.2. Preparation of Dry Cathode Sheet Using Agate Mortar Method (Dry Process)

To prepare the cathode, all materials were dried in a vacuum oven at 80 °C for 3 h. NCM811, CNT or CB, and PTFE were weighed and dispersed using an agate mortar until the color became homogeneous (approximately 1 h). The mixture was spread evenly on a protective film (polyimide film) and compressed using a hydraulic press (30 MPa) for 10 min to reduce voids. Subsequently, it was pressed several times using a roll press to attain the target thickness (200–300 μm) and increase the density of the electrode. The dry cathode sheet was cut into 14 Ø sizes using a punch machine and compressed with 21 µm-aluminum foil at 100 °C using a hydraulic press for 5 min to assemble the coin cell.

### 2.3. Electrochemical Measurements

Using a 14 ∅-sized dry-process cathode, 2032-type half cells were assembled using a polypropylene separator (PP separators film celgard 2400), few drops of 1.2 M LiPF_6_ in EC:EMC (3:7 *v/v* + VC 2 wt.%) as an electrolyte and Li foil (as an anode) under an argon atmosphere glove box (H_2_O < 1 ppm, O_2_ < 0.1 ppm) to evaluate the electrochemical performances of the cathode. After aging, the cycle test was performed in the range of 3.0 to 4.2 V, at a rate of 0.1 C using a battery testing system. The discharge test was conducted at various speeds of 0.1–1 C. The impedance of the coin cells before and after cycling was also evaluated using EIS.

## 3. Results

Figure 1 shows the overall procedure for preparing the cathode using the agate mortar method (dry process without any solvent). Initially, the entire component of the cathode (active materials, carbon conductor, and PTFE) in the powder state was poured and mortared mechanically until a homogenous dispersion was obtained (inset of Figure 1). After sufficient mortaring (approximately 1 h), a paste-like homogeneous mixture was obtained, as shown in Figure 1. Following additional hydraulic pressing and roll pressing to increase the electrode density, these mixtures were placed onto a current collector (aluminum foil) and hot-pressed for 5 min at 100 °C. Additional roll-pressings were performed to ensure enhanced electrode density and a regular interface. Coin cells (2032 type) were then assembled to evaluate the electrochemical performance of the cathode. Further, the effect of the conductor type on the overall battery performance was analyzed.

Figure 2 shows the top and cross-sectional SEM images of the prepared cathodes. Interestingly, in both the CNT- and CB-based cathodes, the spherical shape of NCM811 was maintained after the harsh mortar process. In the cross-sectional SEM images, the average thickness of the dry CNT cathode was 300 μm, and that of the CB-based cathode was 247 μm. No perceptible differences in the top views of the electrodes fabricated using different carbon conductors were evident. However, the electrode prepared using the CNT conductor showed less porosity, resulting in a higher electrode density. It is also proven by the analysis of BET surface area (Appendix A). The BET surface areas of CB- and CNT-based electrodes are measured as 11.25 m^2^/g, and 4.23 m^2^/g, respectively. In all cases, at different electrode compositions (Table 1), the electrode fabricated using the CNT conductor had a higher electrode density than that fabricated using the CB conductor. The electrode density of the CB-based dry cathode sheet (85:10:5) was 3.02 g/cm^3^, while that of the CNT-based dry cathode sheet of the same ratio showed the highest density of 3.40 g/cm^3^. It also summarized the detail information of each electrode in Appendix A. Furthermore, even at other dry-cathode–sheet ratios, the density of the CNT-based cathode was higher than that of the CB-based cathode sheet. This might be attributed to the fibril structure of the CNT and the ensuing entanglement, which are beneficial for increasing the adhesion between the binder and active materials [33]. The EDX mapping of the cathode sheet from the dry process indicates the noticeable dispersion of all components (Co, Ni, Mn, C, and F) (Appendix A), implying that the mortar method is an excellent alternative to the wet-based method (NMP-based slurry) for the preparation of cathode or anode in LiBs.

The mechanical properties of the cathode slurry from the dry process are also suitable for use in LiB electrodes. Figure 3a,b shows the strain–stress curves of the CNT- and CB-based cathodes with respect to different ratios of the components. In both cases, the maximum stress of the CNT paste is higher than that of the CB paste. This is also the result of the lower porosity of the CNT-based paste, and evidently, the entanglement of CNT is beneficial for increasing the mechanical properties of the electrode. In addition, the overall mechanical properties of the cathode paste from the dry process were comparable to those of flexible battery electrodes, as shown in Figure 3c. The folding/unfolding of the electrode does not deteriorate its overall properties, implying that it can be applied in flexible batteries or pouch batteries.

The CNT paste with higher electrode densities and better mechanical properties than CB paste are also advantageous for ameliorating the electrochemical performance of LiB. The coin cell was assembled using Celgard as the separator, Li metal as the anode, and LiPF_6_ in EC:EMC (3:7 *v/v* + VC 2 wt.%) as the electrolyte to analyze the electrochemical performance of the dry-processed cathode. We employed electrodes comprising components of different ratios and observed that the cycle performance in the LiB was enhanced with increasing amounts of carbon conductors in all cases. This indicates that the electron conductance in the cathode is a significant factor in the dry process. However, the recent trend is to increase storage capacitance in LiBs; therefore, determining the optimal condition to utilize a minimum amount of carbon conductor is essential. Fortunately, CNT conductors exhibit better performance than CB conductors at low concentrations. Figure 4a shows a comparison of the cycle performances of the CNT- and CB-based cathodes at a ratio of 85:10:5 (NCM811:carbon:binder). A lower amount of CNT than used in this condition shows poor stability in the cycle test in our manual mortar procedure. Under these conditions, the cycle performance of the CB paste is still poor, but a stable cycle performance can be obtained in the case of CNT at a 85:10:5 ratio until 100 cycles at a charge/discharge rate of 0.1 C. The discharge capacity of the CNT-based cathode was 205 mAh g^−1^, which remained at 155 mAh g^−1^ after 100 cycles with a capacity retention of 75%, whereas the initial capacity of the CB-based cathode cell was 175 mAh g^−1^, which significantly decreased after thirty cycles. The capacity retention of the CNT-based cathode is superior to that of the CB-based cathode at different discharge rates (Figure 4b), indicating that a one-dimensional carbon conductor, such as CNT, is more advantageous than CB in the dry process. General 0-dimensional CB exists as point particles in the electrode mixture, so a large amount is required to connect to each material. On the other hand, since one-dimensional CNT are in the form of linear, it can uniformly connect electrode materials with a small amount, and it helps charges to move better to the electrode than CB. The C-rate capacity of the CNT- and CB-based cathode was compared under the range of 3.0 to 4.2 V by charging at 0.1 C and discharging at 0.11 C. Until a discharge rate of 0.4 C, the capacities of the CNT- and CB-based cathode cells were similar at 94% and 93%, respectively. However, the capacity retention at a higher discharging rate (above 0.6 C) significantly decreased in the case of the CB-based electrode. In particular, when discharged at 1 C, the capacity of the CNT-based cell was 67%, and that of the CB-based cell was 52%, indicating that the CNT-based cell showed better performance than the CB-based cell.

The advantage of the CNT paste in the dry process is more evident in the Nyquist plot of each cathode (Figure 5a,b) after the cycle test. The impedances of CNT and CB were initially comparable (slightly higher in CB paste than in CNT paste), but in the case of the CNT-based cathode, the resistance increased by approximately 1.5 times after 60 cycles, whereas the resistance of the CB-based cathode increased significantly by approximately three times. In the case of the CNT based cathode, the resistance was 16 Ω at 1 kHz and 23 Ω at 1 Hz, and after 60 cycles, each resistance increased to 28 Ω and 49 Ω. The resistance of the CB-based cathode was 18 Ω at 1 kHz and 27 Ω at 1 Hz, which was initially similar to that of CNT. However, after 60 cycles, the resistance of the CB-based cathode more increased to 31 Ω at 1 kHz and 63 Ω at 1 Hz. The impedance loss of the CB-based conductor was much larger than that of the CNT-based conductor, indicating the poor formation of the electrical current pathway during the mortar process. After charge/discharge, the conductivity of the CNT-based paste was maintained, resulting in a minimal decrease in capacity after the cycle, which could also be a reason for the better contact of the CNT with the active materials in the matrix. The SEM images obtained after the cycle test provided direct evidence for the aforementioned results. Figure 5c,d shows the top and cross-sectional SEM images of each electrode, respectively, after the cycle test. In the top view, a gap between the active materials and the binder is more obvious in the case of the CB-based conductor (arrows in Figure 5d, left panel) than that of the CNT-based conductor (Figure 5c, left panel). In addition, these low bonding strengths resulted in the detachment of the active materials from the electrode during the cross-sectioning procedure (Figure 5d, right panel).

## 4. Conclusions

A cathode paste was successfully prepared using a mortar-based dry process. The cathode slurry from the dry process exhibits stable charge–discharge cycle performance when CNT is adopted as carbon conductor. The maximum capacity of the CNT paste was 205 mAh/g, and the capacity retention rate after 100 cycles was 75%, while the highest capacity of the CB paste was 175 mAh/g and significantly decreased after 30 cycles. In the C-rate test (charging at 0.1 C, discharging at 0.1 ~ 1 C), the CNT-based cell showed better performance than the CB-based cell, especially when discharged at 1 C. Furthermore, in the impedance analysis, the CB-based cell showed significantly increased resistance before cycle performance. Thus, the electrochemical performance of the CNT paste was outstanding compared to the CB paste. The maximum capacity of the CNT paste was higher than that of the CB paste. The enhanced performance of the CNT paste from the dry process indicates that a one-dimensional carbon conductor is beneficial for manufacturing a LiB cathode via a dry process. These fundamental results can provide significant information to battery designers regarding solvent-free processes in battery production.

## Figures and Tables

**Figure 1 nanomaterials-13-00324-f001:**
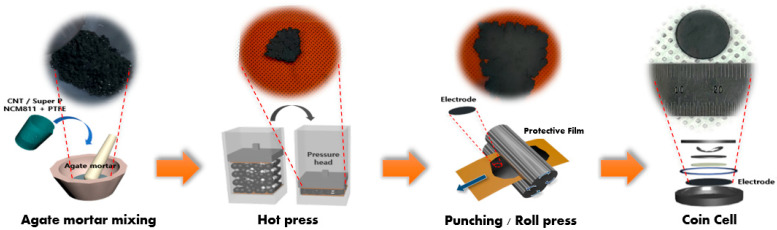
The overall process for dry process LIB cell in this study.

**Figure 2 nanomaterials-13-00324-f002:**
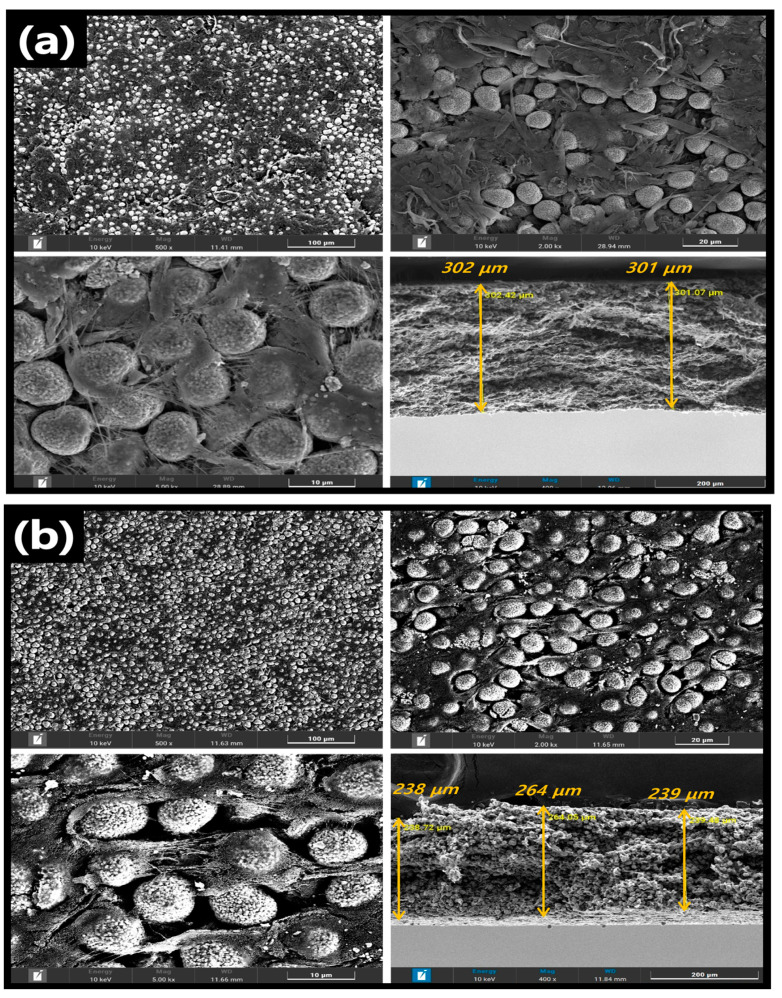
Top and cross−section SEM images of the (**a**) CNT−based cathode and (**b**) CB−based cathode.

**Figure 3 nanomaterials-13-00324-f003:**
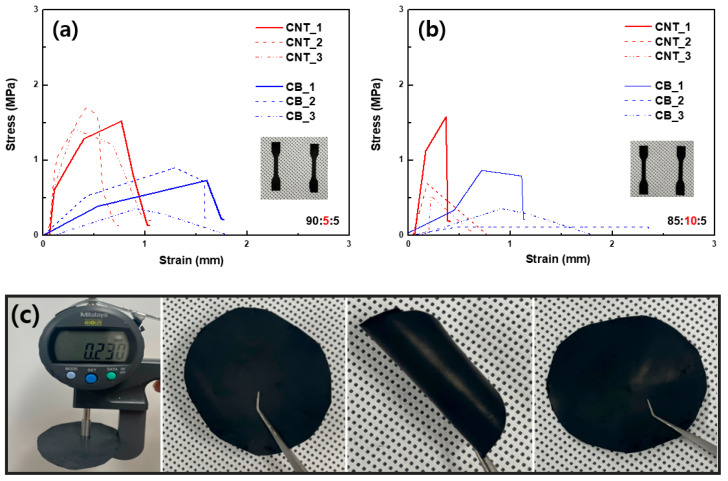
Tensile strength comparison of CNT−and CB−based cathodes from dry process with different compositions of NCM811:carbon:binder: (**a**) 90:5:5 and (**b**) 85:10:5). (**c**) Photo images of a free-standing dry−process cathode sheet before and after folding (CNT-based cathode).

**Figure 4 nanomaterials-13-00324-f004:**
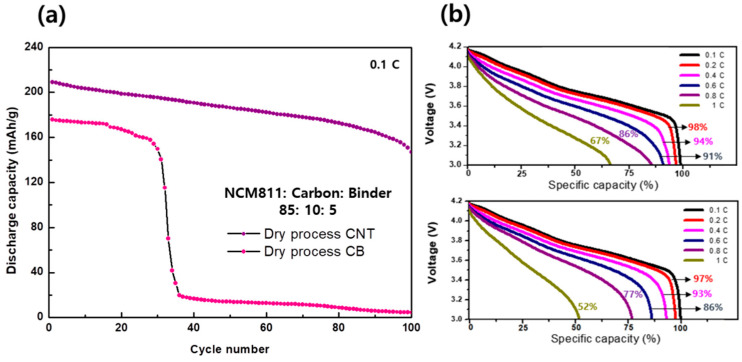
(**a**) Cycling performance of dry-process CNT and CB cathode cell, and (**b**) C−rate test of dry−process CNT and CB cathode cell (charged under 0.1 C rate).

**Figure 5 nanomaterials-13-00324-f005:**
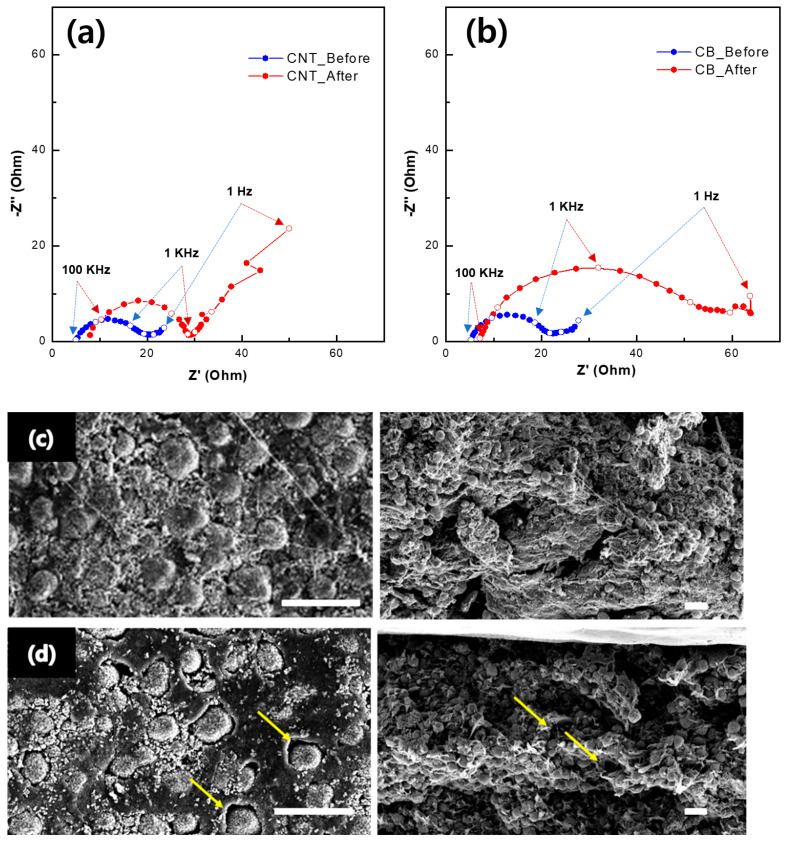
Nyquist plots of before and after cycle performance of (**a**) dry−process CNT cathode cell and (**b**) dry−process Super P cathode cell. Top (**left** column) and cross-sectional SEM images (**right** column) of the dry-process cathode cell after cycle; (**c**) CNT cathode and (**d**) CB cathode (scale bar: 20 μm, Arrow: Gap between the active materials and the binders).

**Table 1 nanomaterials-13-00324-t001:** Electrode density from dry−process CNT and dry−process CB-based cathode sheet.

Sample	Ratio of NCM: Carbon/CB: Binder	Electrode Density (g/cm^3^)	Electrode Loading Level (mg/cm^2^)
Dry-CNT-90	90:5:5	3.25	65
Dry-CNT-85	85:10:5	3.40	71
Dry-CB-90	90:5:5	2.95	65
Dry-CB-85	85:10:5	3.02	60

## Data Availability

Not applicable.

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
