# Peer review of "Solvent-Free Processed Cathode Slurry with Carbon Nanotube Conductors for Li-Ion Batteries"

_nanomaterials, 2023, doi:10.3390/nano13020324_

Round 1
Reviewer 1 Report
Manuscript “Solvent-free processed cathode slurry with carbon nanotube conductors for Lithium-ion batteries” of Gyori Park, Hyun-Suk Kim, Kyung Jin Lee provide procedure for the preparation of electrodes without the use of organic solvent NMP. By this procedure, NCM811 based electrodes with two different types of carbon conductive additives were prepared. CNT based electrode shows better performance in terms of cycling, capacitance retention and mechanical properties of the electrode comparing with CB based one.
My comments and observations regarding this manuscript are as follows:
1. The used materials are not represented in a uniform way:
- for CNT: the manufacturer and the country of the production are indicated,
- for CB: the distributor and content percentage of the main element are indicated,
- for PTFE, brand name, the size of the particles and the density of the material are indicated.
That is, for each material, the output data given is not the same. This does not allow to explain the difference in the density of electrodes based on CNT and CB, respectively.
2. It seems that the drying preparation of materials is insufficient. Commonly , CNT and CB have a large SSA and some porosity.
3. It is possible to suggest that it might be useful to conduct additional experiments to determine the porosity with the BET method.
4. Probably, for the preparation of the mixture for the electrode, it would be possible to use homogenizators or mixers. On the one hand, the process will be standardized, and on the other hand, I think the process time would be reduced significantly.
4. The diameter of the electrode, as far as I understand, is 14 mm. It is useful for evaluating the thickness of electrodes based on various carbon materials.
- for the CNT based electrode (NCM:CNT:PTFE = 85:10:5) - thickness is 300 um (figure 2a),
- for the same electrode, the thickness is estimated at 200 um from values of Table 1.
The difference in the thicknesses of the CB based electrode (NCM:CB:PTFE = 85:10:5) is also observed, although not to such a large extent, as for the CNT based electrode : 240 um (from figure 2b) and 200 um (from table 1), respectively.
Probably, this can explain the different stress values of electrode given on figure 3b. The mechanical properties of the electrodes are difficult to interpret based on the data shown in figure 3b.
5. Lines 92-102, the first paragraph "3. Results" retells paragraph 2.2 and, thus, it is unnecessary. What “enhanced electrode density” means (line 100)?
6. The authors of the manuscript explain better performance of the CNT-based electrode by the fact that it is a 1-dimensional material, but the text does not indicate which dimension CB have? It seems to me that the improvement in the performance of CNT based electrode is not (or not only) related to the 1d dimension of CNTs.
7. Data of the impedance spectroscopy is usually given in the form of square graphs, the axes of which have the same values.
8. Noticed minor descriptions are following :
- line 13 “N-methypyrolidone” instead of “N-methylpyrrolidone”,
- line 16 “MCN811” instead of “NCM811”,
- line 206 "batter" (?),
- the caption of the figure 5 - SEM photographs: “a and b” instead of “c and d”,
- what temperature of hot pressing was : 110oC (line 80) or 100oC (line 99)?
To publish this communication, the authors need to find a more convincing explanation of why CNTs are more efficient when used in an electrode, as well as correct some typos.

Author Response
Response to Reviewer 3’s comments
Manuscript “Solvent-free processed cathode slurry with carbon nanotube conductors for Lithium-ion batteries” of Gyori Park, Hyun-Suk Kim, Kyung Jin Lee provide procedure for the preparation of electrodes without the use of organic solvent NMP. By this procedure, NCM811 based electrodes with two different types of carbon conductive additives were prepared. CNT based electrode shows better performance in terms of cycling, capacitance retention and mechanical properties of the electrode comparing with CB based one.
Answer: Thanks for your great indication! We have tried to incorporate your valuable indications onto the revised manuscript. Thanks again!! In addition, we have done extensive English revisions, as required editor and reviewers!
Comment #1
The used materials are not represented in a uniform way:
- for CNT: the manufacturer and the country of the production are indicated,
- for CB: the distributor and content percentage of the main element are indicated,
- for PTFE, brand name, the size of the particles and the density of the material are indicated.
That is, for each material, the output data given is not the same. This does not allow to explain the difference in the density of electrodes based on CNT and CB, respectively.
Answer: Thank you for your kind advice. We added the detailed information about the materials to be as comparable as possible in 2.1 Materials parts.
“CNT (JEIO, South Korea, 5~7 nm) and Carbon black Super P (Alfa Aesar, Conductive 99+%, 40 μm) was used as the conductive additive materials in this study. LiNi0.8Co0.1Mn0.1O2 (NCM811) (Wellcos, South Korea, 7~14 μm) was used as the cathode materials, and Polytetrafluoroethylene (PTFE) (Chemours-Mitsui Fluoroproducts Co. Ltd, Japan, 495 μm, 482 g/L) was also purchased from Wellcos and used as the binder without further purification.”
Comment #2
It seems that the drying preparation of materials is insufficient. Commonly , CNT and CB have a large SSA and some porosity.
Answer: Yes! It is quite challenging task, and thus several researchers have just begun to adopt dry-process in battery manufacturing. Tedious and enough grinding procedure are now required to reduce void, and high pressure to make those denser is required. We hope that we can present better and optimized procedure to make cathode by dry-process in near future. Several ideas in terms of procedure for dry process or instrumental development are now tested.
Comment #3
It is possible to suggest that it might be useful to conduct additional experiments to determine the porosity with the BET method.
Answer: Thanks for great indication. The BET surface area is additionally analyzed by nitrogen absorption/desorption isotherm. Those of CB and CNT based electrode is measured by 11.25 m2/g, and 4.23 m2/g, respectively.
Following sentences are added in middle of paragraph explaining Figure 2;
“It is also proven by analysis of BET surface area (Figure S1). The BET surface areas of CB and CNT based electrode are measured by 11.25 m2/g, and 4.23 m2/g, respectively.”
Following figures are added in Supporting Figure S1;
Figure S1. BET adsorption and desorption analyses graph of (a) CNT based electrode, (b) CB based electrode
Comment #4
Probably, for the preparation of the mixture for the electrode, it would be possible to use homogenizators or mixers. On the one hand, the process will be standardized, and on the other hand, I think the process time would be reduced significantly.
- The diameter of the electrode, as far as I understand, is 14 mm. It is useful for evaluating the thickness of electrodes based on various carbon materials.
- for the CNT based electrode (NCM:CNT:PTFE = 85:10:5) - thickness is 300 um (figure 2a),
- for the same electrode, the thickness is estimated at 200 um from values of Table 1.
The difference in the thicknesses of the CB based electrode (NCM:CB:PTFE = 85:10:5) is also observed, although not to such a large extent, as for the CNT based electrode : 240 um (from figure 2b) and 200 um (from table 1), respectively.
Probably, this can explain the different stress values of electrode given on figure 3b. The mechanical properties of the electrodes are difficult to interpret based on the data shown in figure 3b.
Answer: Thank you for your kind consideration. In the case of the wet process electrode process, we usually use a homogenizer to disperse conductor or electrode materials. However, since dry processes do not use solvents, we have experienced that agate mortar is more effective than homogenizers and mixers. In addition, wet processing slurry methods require a long process time because the solvent must be sufficiently dried, but the dry process using the PTFE binder by the agate mortar can reduce the processing time since the solvent removal process is not required, if grinding procedure is optimized!
As you said, there was a difference in thickness due to experimental error during preparing the electrode sample for assembling the coin cell and measuring cycle performance. However, in the case of the tensile strength test electrode samples, we prepared dog-bone specimens of the same size as follows picture, and of course, the thickness of each specimen was fixed at 200 um. And also, when we measure the tensile test, we obtained organized data because we feed information about the thickness value of each specimen.
Comment #5
Lines 92-102, the first paragraph "3. Results" retells paragraph 2.2 and, thus, it is unnecessary. What “enhanced electrode density” means (line 100)?
Answer: It is nice indication!! In general, we described brief version of experimental procedure in Result part in order to help easy understanding for general readers. Detailed experimental procedure with quantitative values can be found in experimental part. However, if the reviewers really think this part should be removed, we will delete those out.
About “enhanced electrode density”, we have changed that sentence as following to avoid possible confusion.
“The CNT paste with higher electrode densities and better mechanical properties than CB paste are also advantageous for ameliorating the electrochemical performance of LiB.”
Comment #6
The authors of the manuscript explain better performance of the CNT-based electrode by the fact that it is a 1-dimensional material, but the text does not indicate which dimension CB have? It seems to me that the improvement in the performance of CNT based electrode is not (or not only) related to the 1d dimension of CNTs.
Answer: Thank you for your accurate confirmation. General 0-dimensional CB exists as point particles in the electrode mixture, so a large amount is required to connect to each material. On the other hand, since one-dimensional CNT are in the form of linear, it can uniformly connect electrode materials with a small amount, and it helps charges to move well to the electrode than CB. Of course, as you said, improving the performance of CNT based electrodes is not only related to the one-dimension of CNTs structure, but also related to various experimental parameters. However, here, we are demanding that the difference of dimension is a factor that can have a great effect on the electrochemical performance.
Following sentences are added in revised manuscript (bottom of page 5)
“General 0-dimensional CB exists as point particles in the electrode mixture, so a large amount is required to connect to each material. On the other hand, since one-dimensional CNT are in the form of linear, it can uniformly connect electrode materials with a small amount, and it helps charges to move well to the electrode than CB.”
Comment #7
Data of the impedance spectroscopy is usually given in the form of square graphs, the axes of which have the same values.
Answer: Thank you for your valuable comment. Fig5 a, b is now modified with better quality in the revised manuscript. We rearranged the EIS graph into a square shape and compared the resistance of CNT and CB according to Hz values.
Figure 5. Nyquist plots of before and after cycle performance (a) dry process CNT cathode cell, (b) dry process Super P cathode cell. Top (left column) and cross-sectional SEM images (right column) of the dry process cathode cell after cycle performance
Comment #8
Noticed minor descriptions are following :
- line 13 “N-methypyrolidone” instead of “N-methylpyrrolidone”,
- line 16 “MCN811” instead of “NCM811”,
- line 206 "batter" (?),
- the caption of the figure 5 - SEM photographs: “a and b” instead of “c and d”,
- what temperature of hot pressing was : 110oC (line 80) or 100oC (line 99)?
To publish this communication, the authors need to find a more convincing explanation of why CNTs are more efficient when used in an electrode, as well as correct some typos.
Answer: Thank you for your accurate confirmation. Some typos have been corrected. And the hot pressing temperature was 100 ℃ so it was modified in the manuscript.
Thanks again your helpful and valuable comments!!!
------------------------------------------------------------------------------------------

Reviewer 2 Report
The present manuscript reports the construction of a cathode electrode for a Li-ion battery using NMC811 (LiNi600.8Co0.1Mn0.1O2), PTFE as a binder, and CNT (or CB) as a conductor by dry process. The authors have performed morphological, mechanical, and electrochemical characterization to elucidate the fabricated material. The work is interesting to the readers and the use of a dry process is a different approach; however, the authors haven’t emphasized the novelty of the work. More importantly, the properties of material e.g., structural, chemical, and electrochemical (battery) properties haven’t been analysed and compared with the literature to understand the material truly and the advantage of the dry method. The Li intercalation behaviour to the NMC811 and the reason for poor stability hasn’t been discussed. The current version of the material is not suitable for publication in Nanomaterials.
Minor comments
-MCN811 in a line 16 and NCM811 in a line 47
-The discussions on EIS results and battery performance are missing, the authors should consider fitting the Nyquist plots and compare the results for both CNT and CB.
Author Response
Response to Reviewer 1’s comments
The present manuscript reports the construction of a cathode electrode for a Li-ion battery using NMC811 (LiNi600.8Co0.1Mn0.1O2), PTFE as a binder, and CNT (or CB) as a conductor by dry process. The authors have performed morphological, mechanical, and electrochemical characterization to elucidate the fabricated material. The work is interesting to the readers and the use of a dry process is a different approach; however, the authors haven’t emphasized the novelty of the work. More importantly, the properties of material e.g., structural, chemical, and electrochemical (battery) properties haven’t been analyzed and compared with the literature to understand the material truly and the advantage of the dry method. The Li intercalation behaviour to the NMC811 and the reason for poor stability hasn’t been discussed. The current version of the material is not suitable for publication in Nanomaterials.
Answer: Thanks for your great indication! We have added several sentences to emphasize importance of dry process as you indicated. The NCM 811 and other materials are normally adopted in previous literature, including our previous works (ref 9), so we didn’t add detailed data for material characterization. The poor stability is mainly attributed to the insufficient packing during the dry process, and we are now actively working on this. I hope we can show state-of-art procedure to produce dry-processed cathode in near future. In addition, we have done extensive English revisions, as required editor and reviewers!
Comment #1
Minor comments, MCN811 in a line 16 and NCM811 in a line 47
Answer: Thank you for your accurate confirmation. Every typo have been also corrected throughout the manuscript including a line 16.
Comment #2
The discussions on EIS results and battery performance are missing, the authors should consider fitting the Nyquist plots and compare the results for both CNT and CB.
Answer: Thank you for your valuable comment. Fig 5 a, b is now modified with proper fitting in the revised manuscript as follow. We rearranged the EIS graph into a square shape and compared the resistance of CNT and CB according to Hz values. An EIS analysis part was added to lines 193-197, and the discussion of electrochemical analysis was added to the conclusion part as follows.
Figure 5. Nyquist plots of before and after cycle performance (a) dry process CNT cathode cell, (b) dry process Super P cathode cell. Top (left column) and cross-sectional SEM images (right column) of the dry process cathode cell after cycle performance
“In the case of the CNT based cathode, the resistance was 16 Ω at 1 kHz and 23 Ω at 1 Hz, and after 60 cycles, each resistance was increased to 28 Ω and 49 Ω. The resistance of the CB based cathode was 27 Ω at 1 kHz and 18 Ω at 1 Hz, which is initially similar to that of CNT. However, after 60 cycles, the resistance of CB based cathode more increased to 31 Ω at 1 kHz and 63 Ω at 1 Hz.”
“The maximum capacity of CNT paste was 205 mAh/g and the capacity retention rate after 100 cycles was 75%, while the highest capacity of CB paste was 175 mAh/g and significantly decreased after 30 cycles. In the C-rate test (charging at 0.1 C, discharging at 0.1 ~ 1 C), the CNT based cell showed better performance than CB based cell, especially when discharged at 1 C. Also, in the impedance analysis, CB based cell showed significantly increased resistances before cycle performance. So, the electrochemical performance of CNT paste was outstanding compared to CB paste.” ( Part 4. Conclusions, line 220~227.)
Thanks again your helpful and valuable comments!!!
------------------------------------------------------------------------------------------

Reviewer 3 Report
In this paper, the authors reported the fabrication of cathode electrode based dry process using MCN811, 16 carbon conductor and PTFE binder. Two different types of carbon conductors are compared, and electrode from carbon nanotube shows much better performances . The work is interesting and worth publications after following revisions:
1)In the introduction part, the development of dry process should be briefly introduced to show the advance of the work.
2) The electrolyte for the battery should be presented.
3) From figure4a, it can be seen that the CNT based electrode start to decay after about 100 cycles, more discussion should be added to improve the performance.
4) Related references may be cited, such as DOI:10.1149/2.0081913jes
Author Response
Response to Reviewer 2’s comments
In this paper, the authors reported the fabrication of cathode electrode based dry process using MCN811, 16 carbon conductor and PTFE binder. Two different types of carbon conductors are compared, and electrode from carbon nanotube shows much better performances . The work is interesting and worth publications after following revisions:
Answer: Thanks for your great indication! We have tried to incorporate your valuable indications onto the revised manuscript. Thanks again!! In addition, we have done extensive English revisions, as required editor and reviewers!
Comment #1
In the introduction part, the development of dry process should be briefly introduced to show the advance of the work.
Answer: Thank you for your kind consideration. We add about the advances of the dry process such as the shortening of the procedure and reduction of voids on the electrode surface in the introduction part as follows.
“In addition, wet processing such as NMP and aqueous-based slurry methods require a long process time and cost because the solvent must be sufficiently dried, and microstructural defects may occur on the electrode surface while the solvent evaporated.” (In line 40~43)
“Since the solvent removal process is not required, the dry process using the PTFE binder can reduce the processing time and does not occur unnecessary voids on the electrode surface.” (In line 54~56)
Comment #2
The electrolyte for the battery should be presented.
Answer: Thank you for your comment. The type of the electrolyte we used in our experiments was added to 2.3 Electrochemical measurements. (1.2M LiPF6 in EC : EMC (3 : 7 v/v + VC 2 wt.%))
Comment #3
From figure4a, it can be seen that the CNT based electrode start to decay after about 100 cycles, more discussion should be added to improve the performance.
Answer: It is great indication!! As indicated in manuscript, larger amount of conductor is normally helpful to enhance cycle test even more than 100 cycles in the case of dry process. However, maximum loading of active materials in electrode is also important issue, and thus we decided to exhibit cycle data with 85:10:5 ratio of NCM811:carbon:binder. Of course, the optimization is surely required. In general, higher pressure to make dense packing of electrode is helpful to enhance cycle performance, but too much higher pressure can cause breakage of active materials (NCM).
Comment #4
Related references may be cited, such as DOI:10.1149/2.0081913jes
Answer: Thank you for recommending a good reference. We have added this reference as a number 12 in manuscript.
Thanks again your helpful and valuable comments!!!

Round 2
Reviewer 1 Report
Manuscript “Solvent-free processed cathode slurry with carbon nanotube conductors for Lithium-ion batteries” of Gyori Park, Hyun-Suk Kim, Kyung Jin Lee provide procedure for the preparation of electrodes without the use of organic solvent NMP. By this procedure, NCM811 based electrodes with two different types of carbon conductive additives were prepared. CNT based electrode shows better performance in terms of cycling, capacitance retention and mechanical properties of the electrode comparing with CB based one.
I thank the authors for the corrections made to the work, which, in my opinion, undoubtedly improved the design and presentation of the data.
1. The BET measurement data clearly demonstrate the increased area of CB based electrodes compared to CNT-based electrodes, which may confirm the effect of porosity on the mechanical properties of the prepared electrodes. Could the authors control the porosity of the CB-based electrodes using roll calendering and thus improve the electrodes and increase their performance?
I tried to estimate specific mass loading based on the data of table 1.
If the density of the CNT based electrodes is 3250-3400 mg/cm^3 (table 1), the thickness is 0.03 cm (figure 2a), and electrode area : S = π(D^2/4) = 1.539 cm^2. My estimation of mass loading was 150 - 157 mg/cm^2, what differs from the data of table 1 twice. Approximately the same data I received for CB based electrodes.
Can the authors explain this difference in the received values?
2. Some of the misprints seen in the manuscript are as follows:
- line 14 “N-methypyrolidone” instead of “N-methylpyrrolidone”, double "r"
- line 67 “um ” instead of “nm”,
- line 92 it seems it is more correct “few drops of 1.2M LiPF6 in EC : EMC (3:7 v/v) + VC 2 wt.% as an electrolyte",
- line 155 “85:5:10 ” instead of “85:10:5”,
- line 201 it seems that sentence "The resistance of the CB based cathode was 27 Ω at 1 kHz and 18 Ω at 1 Hz ...." is wrong, or figure 5b presents other data.
I think that the manuscript of communication may be published after the above comments are corrected.

Author Response
Response to Reviewer 1’s comments
Manuscript “Solvent-free processed cathode slurry with carbon nanotube conductors for Lithium-ion batteries” of Gyori Park, Hyun-Suk Kim, Kyung Jin Lee provide procedure for the preparation of electrodes without the use of organic solvent NMP. By this procedure, NCM811 based electrodes with two different types of carbon conductive additives were prepared. CNT based electrode shows better performance in terms of cycling, capacitance retention and mechanical properties of the electrode comparing with CB based one.
I thank the authors for the corrections made to the work, which, in my opinion, undoubtedly improved the design and presentation of the data.
Answer: Thanks for your kind indications and considerations on our manuscript. We have tried to revise manuscript base on your nice indications!
Comment #1
The BET measurement data clearly demonstrate the increased area of CB based electrodes compared to CNT-based electrodes, which may confirm the effect of porosity on the mechanical properties of the prepared electrodes. Could the authors control the porosity of the CB-based electrodes using roll calendering and thus improve the electrodes and increase their performance?
I tried to estimate specific mass loading based on the data of table 1.
If the density of the CNT based electrodes is 3250-3400 mg/cm^3 (table 1), the thickness is 0.03 cm (figure 2a), and electrode area : S = π(D^2/4) = 1.539 cm^2). My estimation of mass loading was 150 - 157 mg/cm^2, what differs from the data of table 1 twice. Approximately the same data I received for CB based electrodes.
Can the authors explain this difference in the received values?
Answer: Thank you for your valuable comment.
Yes, it is right! During the electrode manufacturing process, both CNT based electrode and CB based electrode rolled by the same percentage in this study (About 80% of the initial thickness), and under the same pressing condition, performance of CNT based electrode is higher than that of CB. However, as long as the NCM811 cathode material is not broken, we think the porosity or density can be adjustable.
In the case of loading amount, there was a difference in thickness due to experimental error during preparing the electrode sample, so when we calculated the electrode density and loading level, we calculated the average thickness and weight values of each type of electrode. For better understanding, we summarized the detail information of each electrode in Table S1. In the case of CNT based 90:5:5 electrode, the electrode area is 1.54 cm2 (S = π(D^2/4) = 1.539 cm2), and the electrode density is 3.25 (Weight/(Thickness*Electrode area)=3.2467 g/cm3), and the electrode loading level is 65 (Weight/Electrode area=64.935 mg/cm2).
Table S1. Detail information from Dry process CNT, Dry process CB based cathode sheet
|
Ratio of NCM: Carbon/CB: binder |
Electrode area (cm2) |
Thickness (μm) (Without Al foil) |
Weight (g) (Without Al foil) |
Electro density (g/cm3) |
Electrode Loading Level (mg/cm2) |
|
Dry-CNT (NCM811 : CNT : PTFE) |
|||||
|
90 : 5 : 5 |
1.54 |
200 |
0.1 |
3.25 |
65 |
|
85 : 10 : 5 |
1.54 |
210 |
0.11 |
3.4 |
71 |
|
Dry-CB (NCM811 : CB : PTFE) |
|||||
|
90 : 5 : 5 |
1.54 |
220 |
0.1 |
2.95 |
65 |
|
85 : 10 : 5 |
1.54 |
200 |
0.093 |
3.02 |
60 |
Comment #2
Some of the misprints seen in the manuscript are as follows:
- line 14 “N-methypyrolidone” instead of “N-methylpyrrolidone”, double "r"
- line 67 “um ” instead of “nm”,
- line 92 it seems it is more correct “few drops of 1.2M LiPF6 in EC : EMC (3:7 v/v) + VC 2 wt.% as an electrolyte",
- line 155 “85:5:10 ” instead of “85:10:5”,
- line 201 it seems that sentence "The resistance of the CB based cathode was 27 Ω at 1 kHz and 18 Ω at 1 Hz ...." is wrong, or figure 5b presents other data.
I think that the manuscript of communication may be published after the above comments are corrected.
Answer: Thank you for your accurate confirmation! Some typos have been corrected.
Thanks again your helpful and valuable comments!!!

Reviewer 2 Report
The authors have considered the reviewer`s comments and amended the manuscript accordingly.
Author Response
Response to Reviewer 2’s comments
The authors have considered the reviewer`s comments and amended the manuscript accordingly.
Answer: Thank you for kind indications and considerations on our manuscript.
Thanks again for valuable comments and indications!!!